# Designing solid-liquid interphases for sodium batteries

Snehashis Choudhury[1], Shuya Wei[1], Yalcin Ozhabes[2], Deniz Gunceler[2], Michael J. Zachman[3], Zhengyuan Tu[4], Jung Hwan Shin[1], Pooja Nath[1], Akanksha Agrawal[1], Lena F. Kourkoutis[3,5], Tomas A. Arias[2] & Lynden A. Archer[1]

Secondary batteries based on earth-abundant sodium metal anodes are desirable for both stationary and portable electrical energy storage. Room-temperature sodium metal batteries are impractical today because morphological instability during recharge drives rough, dendritic electrodeposition. Chemical instability of liquid electrolytes also leads to premature cell failure as a result of parasitic reactions with the anode. Here we use joint density-functional theoretical analysis to show that the surface diffusion barrier for sodium ion transport is a sensitive function of the chemistry of solid–electrolyte interphase. In particular, we find that a sodium bromide interphase presents an exceptionally low energy barrier to ion transport, comparable to that of metallic magnesium. We evaluate this prediction by means of electrochemical measurements and direct visualization studies. These experiments reveal an approximately three-fold reduction in activation energy for ion transport at a sodium bromide interphase. Direct visualization of sodium electrodeposition confirms large improvements in stability of sodium deposition at sodium bromide-rich interphases.

[1] School of Chemical and Biomolecular Engineering, Cornell University, Ithaca, NY 14853, USA. [2] Department of Physics, Cornell University, Ithaca, NY 14853, USA. [3] School of Applied and Engineering Physics, Cornell University, Ithaca, NY 14853, USA. [4] Department of Materials Science and Engineering, Cornell University, Ithaca, NY 14853, USA. [5] Kavli Institute at Cornell for Nanoscale Science, Cornell University, Ithaca, NY 14853, USA. Correspondence and requests for materials should be addressed to L.A.A. (email: laa25@cornell.edu)

Rechargeable batteries based on lithium and sodium metal anodes are of interest for high-energy storage solutions in portable and stationary applications[1, 2]. Although sodium-based batteries pre-date those based on lithium[3], Li has received more recent attention for a variety of reasons, including its greater electronegativity, higher specific energy, low atomic radius[4, 5], and the commercial success of related Li-ion battery technology. The greater natural abundance of sodium and its availability in regions all over the world provide significant cost advantages over Li that have within the last decade helped re-ignite interest in Na-based batteries[6–8]. Metallic sodium has other attractive features as a battery anode, including its relatively high electronegativity and low atomic weight, which combine to give the Na anode a specific capacity (1166 mAh gm$^{-1}$) that is competitive with Li (3860 mAh gm$^{-1}$) in many applications[6]. Additionally, recent studies have shown that rechargeable batteries that pair a Na anode with highly energetic $O_2$-based cathodes are intrinsically more stable during discharge than their Li analogs because the species generated electrochemically in the cathode, the metal superoxide, is more stable when the anode is Na, as opposed to Li[9, 10].

As with rechargeable batteries comprising Li metal anodes, the Achilles heel of the rechargeable sodium battery is the anode's susceptibility to failure during the charging process. Specifically, during battery recharge Na ions deposit in rough, low density and uneven patches on the negative electrode, even at current densities below the limiting current where classical instabilities such as electroconvection that drive rough, dendritic deposition are expected to be unimportant[11, 12]. Instead, dendrites on Na (and Li) arise from inhomogeneities in the resistance of the solid–electrolyte interphase (SEI), formed spontaneously on the anode surface when in contact with an electrolyte. The resultant concentration of electric field lines on faster growing regions of the interface drives the morphological instability loosely termed dendrites[12, 13]. At later stages, uncontrolled dendritic deposition leads to metallic structures able to bridge the inter-electrode space, ultimately short-circuiting the cell. Short-circuits lead to two catastrophic failure mechanisms: (i) Thermal runaway that drives chemical reactions in the electrolyte, ending the cell life by fire, explosion or both[12, 14–16]; and (ii) Melting and breakage of the dendrites, which electrically disconnects the material from the electrode mass[4, 17], causing rapid or gradual reduction in the storage capacity of the anode. Unlike Li, where dendrite-induced short circuits are considered the dominant failure mode, chemical reaction between the electrolyte and metal anode are regarded as the most important mechanism of cell failure for batteries based on a Na anode. Na also has a lower melting point than Li, which makes batteries based on Na more prone than their Li counterparts to failure by thermal runaway and/or dendrite breakage[6, 18, 19].

Few studies have addressed the challenges associated with stabilizing a Na anode[18]. In contrast, several approaches have been reported for preventing/retarding Li dendrite proliferation in Li metal batteries[11, 12]. Some of the approaches include using high modulus electrolyte or nanoporous/tortuous separator[14, 20–22], modifying the ion transport in electrolytes by using single ion conductors and ionic liquids[23–27], or forming a stable electrode-electrolyte interface to suppress the nucleation of dendrites[4, 13, 28–30]. In addition to preventing dendrite induced short circuits, the last approach may impede unwanted parasitic reactions between the electrode and electrolyte that lead to formation of insulating products and loss of electrochemically active material, causing decay in the battery capacity with increasing charge-discharge cycles[12]. A common approach for the formation of artificial SEI on the metal involves use of special electrolyte additives such as vinylene carbonate[31, 32], fluoroethylene carbonate[33], dioxane[34], sultones[30, 35], or functional ionic liquids[7], which can electro-polymerize on the surface of electrode to form an elastic coating that protects the metal surface and accommodate volume changes in the electrode during charge and discharge. There are also recent reports of protecting the electrode interface by direct formation of a barrier layer by deliberate reaction between electrodes and reactive species in electrolytes[36–38]. Various indirect methods have also been reported for stabilizing a Li anode during battery recharge. These include use of a functional nanoparticles[6, 11, 23, 24], and mixtures of salts (e.g. LiTFSI-LiFSI)[39], use of concentrated electrolytes (e.g. 5 M LiFSI in DME)[40], or use of polysulfides and LiNO$_3$[29] as electrolyte additives. A common feature of these methods is that they produce lithium fluoride (LiF) in the SEI. In recent studies[13, 22], direct incorporation of LiF as an additive in liquid electrolytes was reported to yield dramatic enhancements to battery lifetime in Li metal cells at both high and low current densities. Cui and co-workers[18] were among the first to show that application of this concept to Na metal batteries, through electrolyte additives that generate sodium fluoride (NaF), leads to markedly higher coulombic efficiencies (as high as 99%) in Na‖Cu cells.

With the specific aim of developing rational strategies for stabilizing the anode of Na batteries during cell recharge, we herein investigate how the chemistry of the SEI alters ion transport at Na- and Li-electrolyte interfaces by means of joint density-functional theory (JDFT) calculations and experiment. Our focus on interfacial transport derives from the observation that magnesium metal anodes, which do not form uneven deposits under charging at currents below the limiting current[41], present the lowest barriers for interfacial metal ion transport[42]. Remarkably, we find that a Na metal anode protected by NaBr presents a barrier of only ~ 0.02 eV per atom (i.e. comparable to Mg metal) for interfacial ion transport. By means of direct visualization studies and electrochemical analysis, we investigate the stability imparted to the Na electrode by a NaBr protective coating.

## Results

**Joint density-functional theory study of SEI**. It has been argued that the concentration of electric field lines at protrusions on the electrode surface leads to non-uniform ion distribution and deposition rate, which serves to seed dendrites[43]. We have previously proposed through density-functional simulations that enhanced surface diffusion at the electrode-electrolyte interface could serve as a counter mechanism by smoothing protrusions on the surface and thus prevent formation of dendrites[44–46].

To understand these effects in the context of the sodium anode, we simulated sodium adatoms on the surface of different passivated sodium electrodes using a similar methodology as described in detail in our previous work focused on density-functional calculations of transport barriers for halogenated SEI salt layers in lithium-metal batteries[44]. The surface diffusion barrier is affected by the presence of the liquid electrolyte at the interface and its calculation is thus non-trivial. Regular DFT can provide the total energy of a given configuration (or snapshot) at fixed atomic positions, but to accurately compute the free energy of a solid–liquid interface one must also sample the configuration space of the liquid. While this can be done by molecular-dynamics methods (e.g., QM/MM), such calculations are computationally demanding and to-date haven't been reported for the systems of interest. JDFT[45], which works with thermodynamic averages of the fluid variables, provides an economical alternative to molecular dynamics and provides direct access to free energies without the need for sampling.

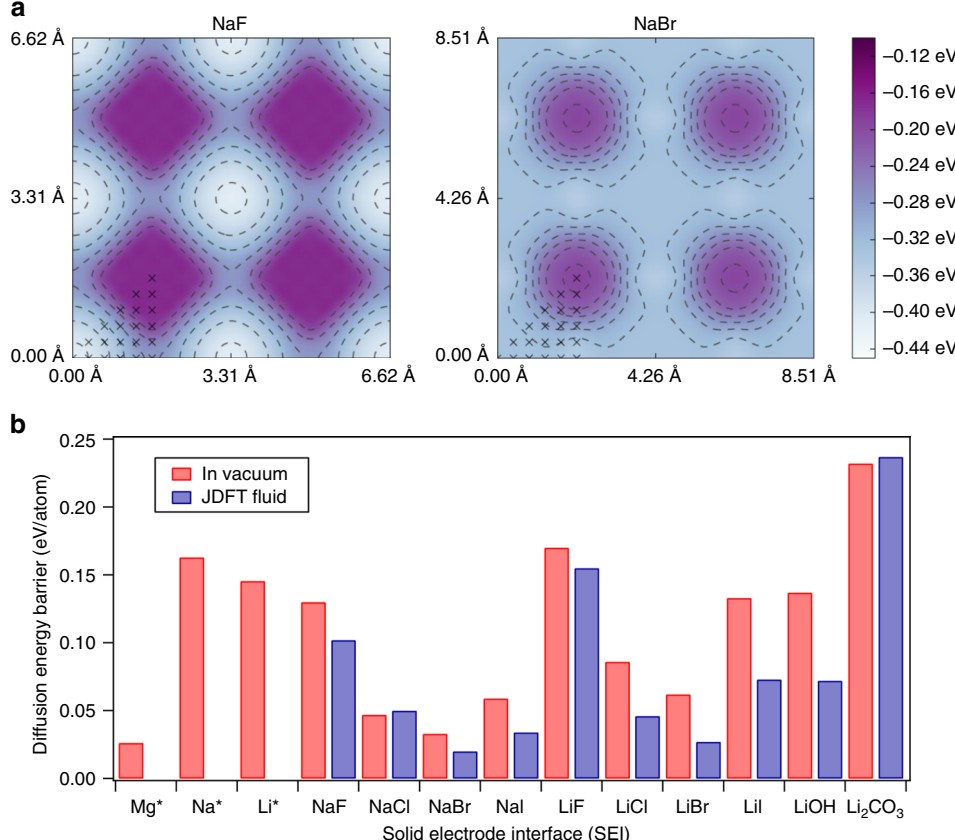

**Fig. 1** Surface diffusion barriers calculated using joint density functional theory. **a** Surface binding energy vs. binding site for NaF (*left*) and NaBr (*right*) obtained from JDFT analysis of adatom diffusion. The entire contour plot is generated by symmetry using the data points indicated by *cross symbols*. **b** Diffusion energy barriers computed for Mg, Na, and Li adatoms on surfaces with the chemistries noted. The *red bars* denote surface in contact with vacuum and *blue bars* indicate the same in presence of acetonitrile. The * symbol marks the data points obtained from ref. [42]

We performed all electronic structure calculations with JDFTx[46], an open-source implementation of JDFT. To account for the effect of the electrolyte, we used the nonlinear polarizable continuum model[45] generalized to non-aqueous solvents[47], which is taken to be acetonitrile in the present work. All calculations employ a plane-wave basis with a cutoff of 20 Hartrees, and we use ultrasoft pseudopotentials from GBRV library[48].

Results for these calculations are reported in Fig. 1. The binding energy of a Na adatom depends on where it binds onto the Na-halide surface (as shown in Fig. 1a). For the smallest halide, F, the minimum energy position for the sodium adatom is directly on top of the fluoride-ion, we refer this site as the "anion site". Again, for F the saddle point on the diffusion path is in the middle of two neighboring anion sites, we call this middle point the "in-between site". Traversing down the periodic table, it is observed that with increasing anion size, the binding energy of the in-between-site becomes relatively closer to the anion-site, and eventually the saddle point becomes the minimum. This transition happens with NaBr and results in the lowest diffusion barrier for interfacial ion transport.

Comparing the surface diffusion barriers of NaBr with other sodium halide salts and lithium salts as well as pure elements, lithium, sodium and magnesium (Fig. 1b) places our finding in perspective with recent theory-supported strategies for suppressing dendritic deposition at metal electrodes. It is notable that the diffusion barrier for NaBr adatoms is substantially lower than for NaF, and even in a liquid electrolyte is comparable to those computed for Mg in vacuum. On the basis of earlier reports that LiF coatings on Li metal dramatically stabilize electrodeposition

of Li[13, 22], and that NaF coatings on Na has a similar large-stabilizing effect on Na deposition[18], we hypothesize that a Na anode protected by a coating of NaBr would be particularly attractive for room-temperature sodium batteries employing liquid electrolytes.

**Formulation and stability of a NaBr-based SEI layer on sodium metal**. To evaluate the JDFT prediction we first developed a method for uniformly coating NaBr on a Na metal electrode. Unlike previous experiments, where the source of halide salts in SEI layer of anode is degradation of active materials[18, 39, 40] or precipitation of a poorly soluble electrolyte salt additive[13, 22], we here employ a well-known chemical reaction to create a layer of NaBr at the interface. Specifically, we carried out a reaction of the sodium metal anode with bromopropane to undergo Wurtz reaction as illustrated in the Fig. 2a. This reaction is widely used for production of symmetric alkanes, with the side product being a sodium halide. In the present case, along with NaBr, hexane is formed, which is removed by evaporation. X-ray diffraction (XRD) analysis (Fig. 2b) for pristine and treated sodium metal confirms that crystalline NaBr is formed on the surface of the Na electrode surface. The morphology of the NaBr layer was interrogated using survey scanning electron microscopy (SEM) for different exposure times of sodium metal piece with 1-bromopropane. This exposure time corresponds to the time the sodium metal piece was dipped into the 1-bromopropane liquid (which we define that as the nominal reaction time, though the actual time of reaction is difficult to quantify). Figure 2c, f shows

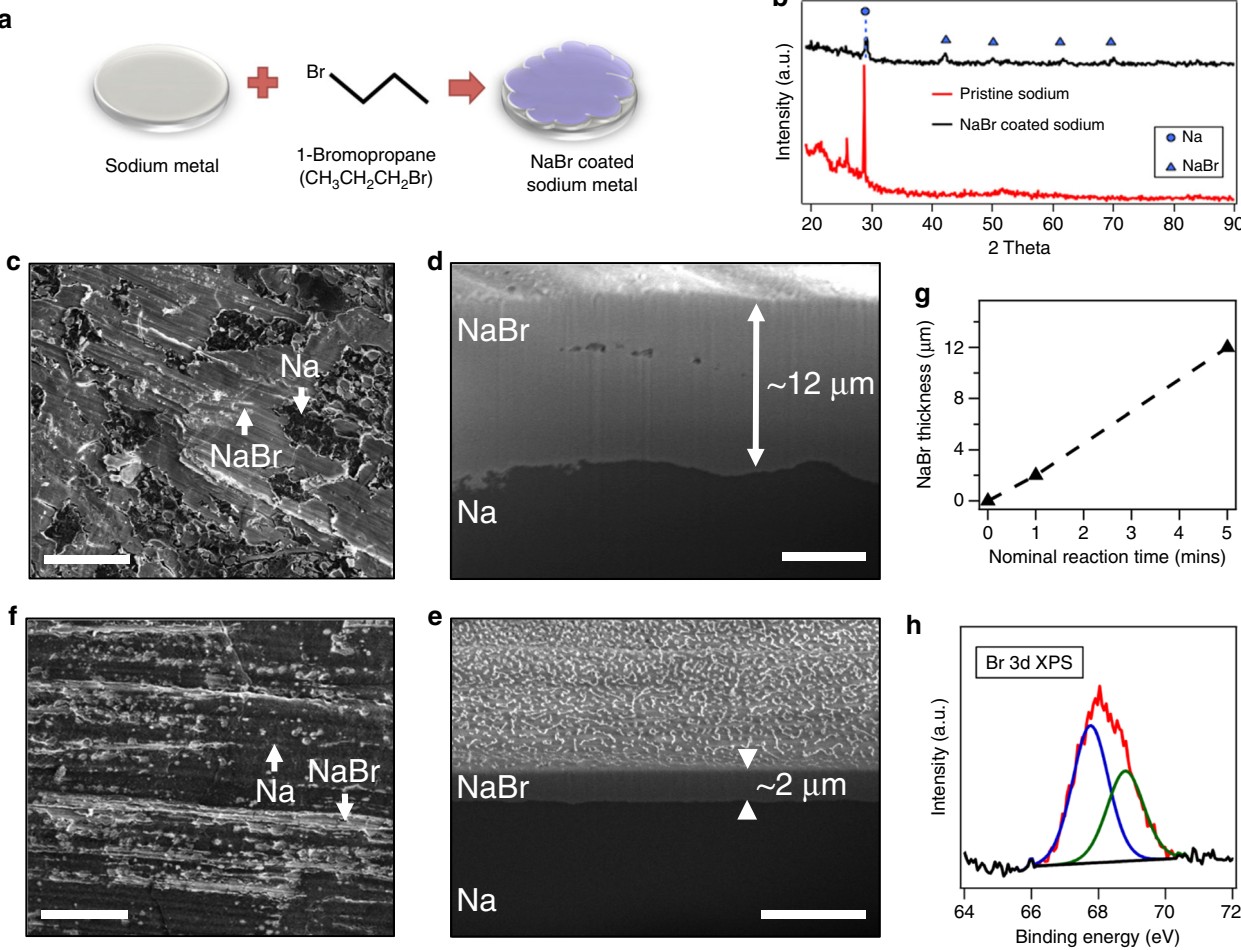

**Fig. 2** Formulation and characterization of sodium bromide layer. **a** Schematic showing the procedure used to coat Na with NaBr. **b** XRD analysis of pristine and NaBr-coated sodium showing that NaBr exists in crystalline form in the coatings on Na. **c** Cryo-SEM image of ~12 μm thick NaBr coating on sodium metal surface. *Light* regions are NaBr coating and *dark* regions are sodium metal, as confirmed by EDX; *scale bar*, 200 μm. **d** Cryo-SEM image of a cross section through the NaBr coating obtained by focused ion beam milling under cryogenic conditions. Cross-sectional SEM imaging was used for layer thickness analysis, and the layer composition was confirmed by EDX mapping; *scale bar*, 5 μm. **e**, **f** Complementary images and *scale bars* as (**d**, **c**), respectively, but for a Na substrate exposed to 1-bromopropane for 1 min. A thinner (2 μm) NaBr coating is formed in this case. **g** Thickness of the NaBr-based SEI layer on sodium metal at various nominal reaction times. **h** X-ray photoelectron spectrum centered on the Br 3*d* bands confirms the existence of metallic bromide bond

the SEM images of the sample surface for exposure times of 5 min and 1 min, respectively. Figure 2c clearly shows large regions of a dense, smooth deposit of NaBr on Na that is interspersed with smaller, less well-coated regions as confirmed by energy dispersive X-ray (EDX). In contrast, fewer well-coated regions are seen in Fig. 2f. More in-depth information about the Na electrode coatings was obtained using cryo-focused ion beam-SEM (cryo-FIB-SEM) and the results are shown in Fig. 2d, e after 5 and 1 mins of treatment of the electrodes with 1-bromopropane, respectively. The thicknesses and depth-dependent composition of the coating layers were determined by SEM imaging and EDX mapping (Supplementary Fig. 1a) of cross sections produced by FIB milling. The EDX element mapping confirms that the top layer is predominantly Br, while the bottom comprises of essentially pure Na metal. The thickness of the NaBr layer is plotted as a function of nominal reaction time in Fig. 2g. For further studies, NaBr-coated sodium samples with nominal reaction times between 1 mins were utilized. X-ray photoelectron spectroscopy (XPS) of the Br 3*d* peaks was used to more carefully analyze the bromine containing compounds in contact with the sodium metal surface. A high-resolution scan was performed after 45 s

sputtering to remove any oxide layer that may form when transferring samples to the XPS chamber. As shown in Fig. 2h, two deconvoluted peaks for Br 3*d* 5/2 and 3/2 at 68.8 eV and 67.7 eV respectively are observed, which correspond to the predominant presence of metallic-bromide bonds on the sodium surface[49–51].

The effectiveness of the NaBr coating in protecting sodium can be most easily evaluated by comparing SEM images of the pristine (Fig. 3a) and NaBr-protected (Fig. 3b) Na electrodes following brief air exposure during transfer to FIB/SEM chamber at room temperature. The former is seen to be covered with a porous oxide layer, which is entirely absent from the NaBr-coated Na. The oxide layer on the pristine sodium surface was ~5 μm thick, as seen from the cross-sectional image in Supplementary Fig. 1b. The stability of the NaBr interphase layer during electrochemical cycling was characterized by post-mortem analysis of the Na anode after five cycles of charge and discharge in a symmetric cell at a fixed current density of 0.5 mA cm$^{-2}$. Figure 3c shows that the NaBr crystal structure is retained, confirming that the anode-protection mechanism is sustained. Results from SEM analysis of the cycled anodes are reported in Fig. 3d. The surface morphology

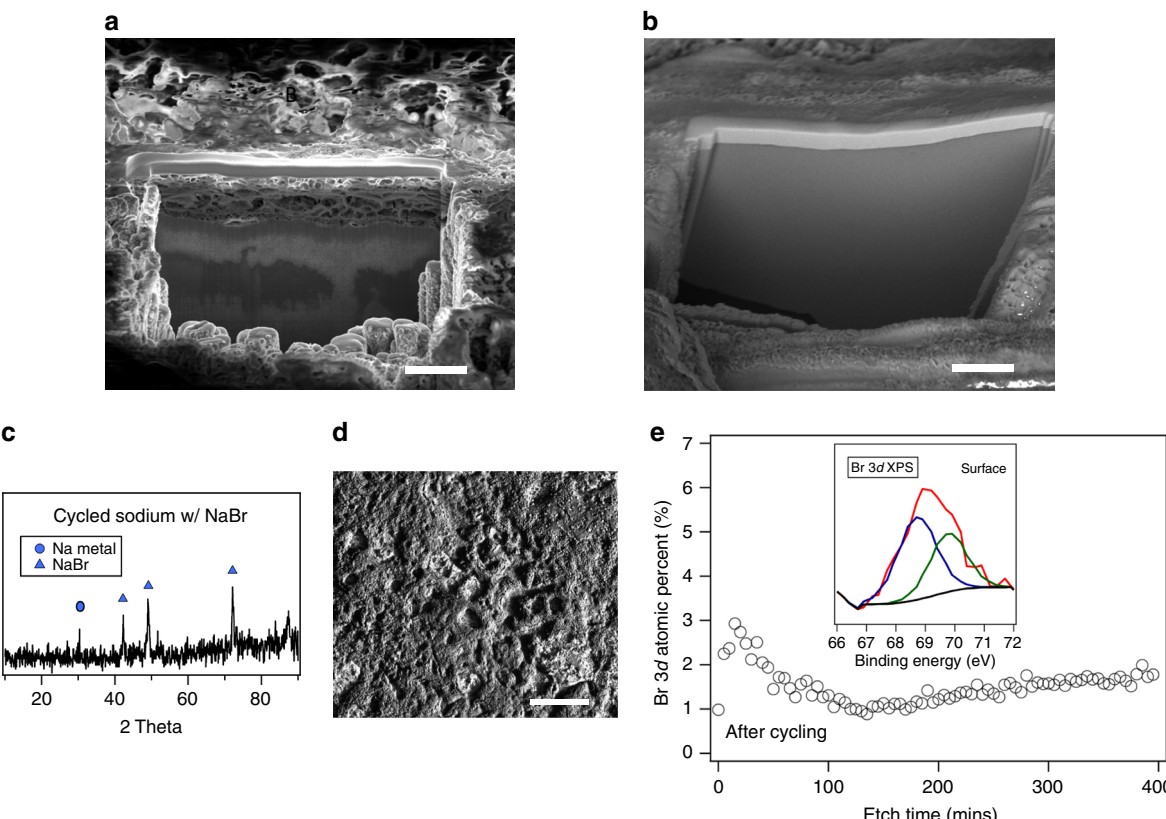

**Fig. 3** Air sensitivity and stability of sodium bromide interphase. **a** Milled region on pristine sodium metal obtained using focused ion beam milling at room temperature. The porous layer indicates severe oxidation of sodium during sample transfer; scale bar, 5 μm. **b** Same as **a** but with NaBr coating. In both (**a**, **b**) the thin top solid layer is a platinum protective coating, deposited inside the FIB prior to milling. **c** XRD showing intensity of Na metal and NaBr peaks for sodium anode with NaBr coating and after cycling at 0.5 mA cm$^{-2}$ for five times in a symmetric sodium cell. **d** SEM image of NaBr-coated sodium metal anode after cycling; scale bar, 200 μm. **e** Depth profiling of cycled anode obtained by ion etching and XPS measurements. The atomic content of Br 3$d$ is shown as a function of etch time. The etching rate is ~5nm/min. The inset shows the Br 3$d$ XPS result for the cycled sodium anode surface (before etching)

is seen to remain relatively flat and compact. XPS analysis for the Br 3$d$ peaks (*inset* of Fig. 3e) was further performed on the sodium sample with NaBr coating after cycling. The depth profile was obtained by etching the surface at 2 kV, 2 μA over an area of 2 mm × 3 mm, at a rate of 5 nm min$^{-1}$ for 395 mins. It is seen from Fig. 3e that the Br 3$d$ atomic content decreases for the first 167 mins, followed by a steady state Br 3$d$ atomic content; this indicates that at least 2 μm thick NaBr layer is retained even after cycling. The surface atomic composition of the cycled sample is deduced from both EDX mapping and XPS analysis (prior to etching) as seen in Supplementary Fig. 2. In both cases, the Br element is seen to co-exist with other elements (phosphorus, fluorine, carbon, oxygen), typical for degradation of the EC/PC NaPF$_6$ electrolyte. It is also seen that with the exception of carbon, the atomic compositions deduced from the two techniques are comparable.

Quantitative assessment of interfacial transport of Na ions in sodium halide coatings was made using impedance spectroscopy. These experiments were performed using symmetric sodium cells with/without halide salt coatings on Na and, for comparison, symmetric magnesium cells. Fig. 4a, b reports Nyquist plots at different temperatures for pristine sodium and NaBr-coated sodium metal symmetric cells, respectively. By fitting the Nyquist plots to an equivalent circuit model (Supplementary Fig. 3) it is possible to deduce from the data the bulk resistance ($R_b$), representing ion transport in the electrolyte, and two interfacial resistances ($R_{int1}$ and $R_{int2}$) representing ion transport through

the passivating layer on Na as well as electronic transport. The temperature dependence of the interfacial ion conductivity can be used to extract information about how the halide coating alters the energy barrier for transport. The reciprocal of bulk impedance and net interfacial resistance are plotted with temperature in Arrhenius form as in Fig. 4c. The temperature-dependent analogs of these plots for NaCl-, NaI-coated sodium as well as that of Mg are provided in the Supplementary Fig. 4. The lines through the data in Fig. 4c are fits obtained using the Vogel—Fulcher—Tammann (VFT) formula[1], $\sigma = A\,\exp(-E_a/R(T-T_o))$, commonly used for modeling ion transport in liquid electrolytes. Here $A$ is the prefactor, $E_a$ is the apparent activation energy for ion transport, $R$ is the universal gas constant and $T_o$ is the reference temperature. The respective VFT coefficients for all materials used in the study are tabulated in Supplementary Table 1. It is seen that the bulk impedance for sodium cells utilizing pristine and halide-coated sodium are similar, indicating that such coatings have at best a minimal effect on ion transport in the liquid electrolyte. In contrast, the interfacial conductivity and its temperature dependence are seen to be very sensitive to the chemistry of the SEI. Figure 4c for example shows that whereas $1/R_{int1}$ for pristine Na is higher than for the NaBr-coated material, it decreases more rapidly with temperature. This latter behavior can be captured in terms of the apparent activation energy $E_a$ for interfacial ion transport, which is reported in Fig. 4d for various pristine and halide-coated sodium electrodes, as well as for Mg. It is observed that $E_a$ for the pristine sodium

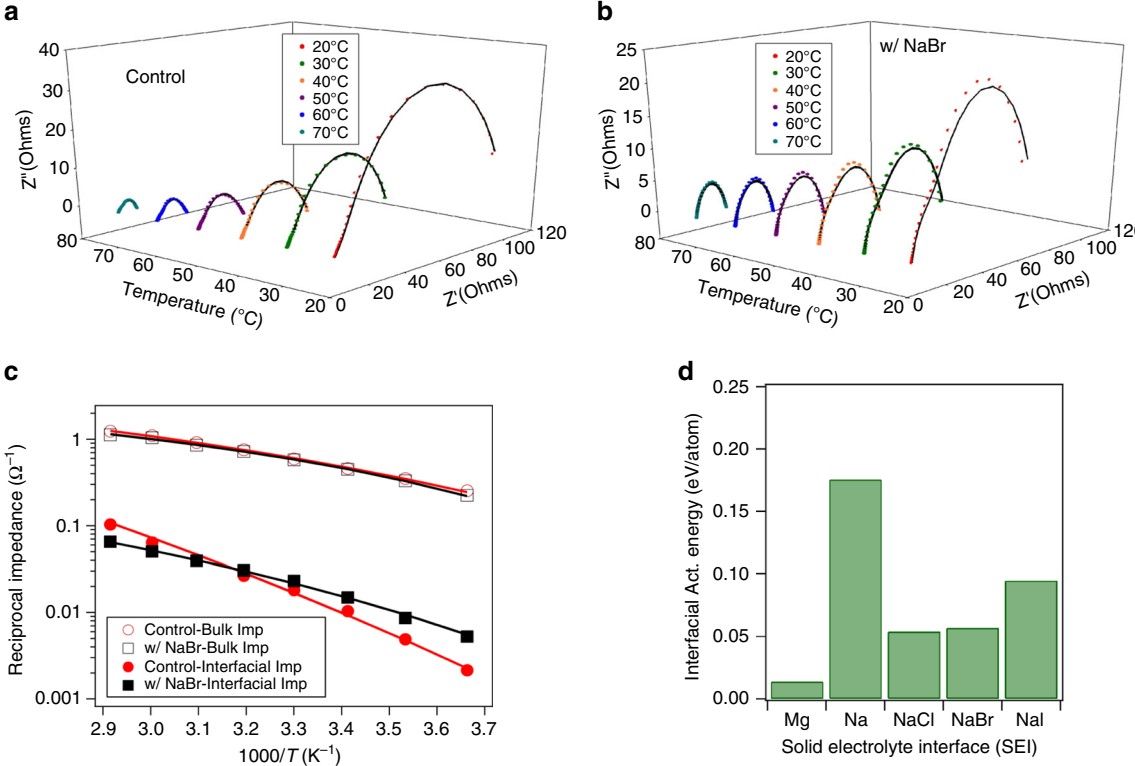

**Fig. 4** Electrochemical impedance spectroscopy analysis temperature-dependent Nyquist plots for a symmetric sodium cell with (**a**) pristine sodium and (**b**) NaBr-coated sodium. **c** Reciprocal bulk and interface impedance as a function of reciprocal temperature; the lines are VFT model fits. **d** The apparent interfacial activation energy obtained from VFT fits of the temperature-dependent reciprocal resistance for various interphase chemistries. Here, Na and Mg symbolize results from symmetric cell studies with pristine sodium and magnesium electrodes, while NaCl, NaBr, NaI are for the corresponding salt-coated Na electrodes

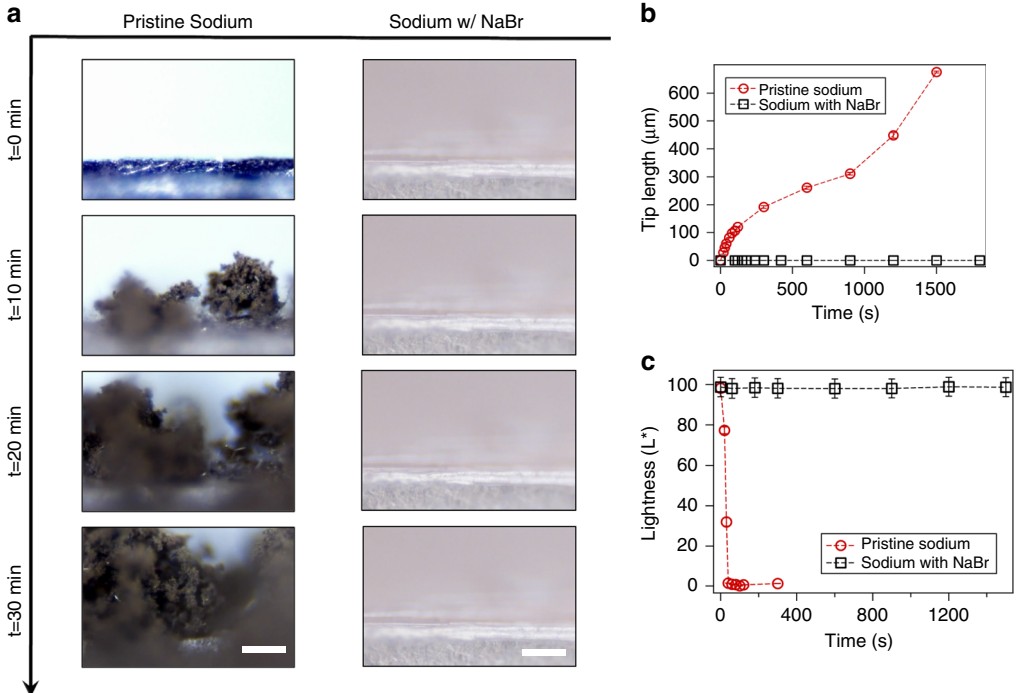

**Fig. 5** Visualization of sodium electrodeposition. **a** Snapshots from video microscopy of the pristine Na- (*left* column) and NaBr-coated Na- (*right* column) electrolyte interface at 0, 10, 20 and 30 min (*top* to *bottom*) after onset of Na deposition at $J = 1\,mA\,cm^{-2}$; *scale bar*, 100 μm. **b** Na dendrite tip length as a function of time. **c** Relative brightness (L*) of electrolyte near the electrode-electrolyte interface as a function of time. This variable captures the obvious darkening of the Na deposits in the pristine case and the complete absence of this effect for the NaBr-coated electrodes. In parts (**b**, **c**), the error bars represent the standard deviation of measurements taken at different points in the same image

(~ 0.175 eV atom$^{-1}$) is higher by a factor of around 3 than the corresponding NaBr- or NaCl-coated metal. This means that at any temperature transport of Na ions is 20-times or more faster in a SEI composed of NaBr or NaCl, in comparison to the SEI formed spontaneously at the pristine Na electrode. These experiments also reveal that the apparent interfacial activation energy for the Mg-symmetric cell (~ 0.02 eV atom$^{-1}$) is around 10-times lower than for pristine Na, although the value of interfacial resistance for Mg is two orders of magnitude higher. As illustrated in Supplementary Fig. 5, these conclusions are broadly insensitive to the model (VFT or Arrhenius) used to extract $E_a$ from the temperature-dependent electrochemical impedance measurements.

A lower $E_a$ for a halide-rich SEI on Na means that at any current density deposition of the ions at the Na interface is less restricted. This result is consistent with the earlier prediction based on our JDFT analysis and is interpreted here to mean that a low diffusion energy barrier for halide adatom transport contributes to the low interfacial activation energy measured here. Additionally, the fact that we experimentally capture both the trends with Na-halide salts and the dramatically lower $E_a$ values for Mg-electrolyte interphases predicted by the JDFT analysis implies that the diffusion barrier to adatom transport

dominates the overall energy barrier to ion transport at the SEI. Comparison of the $E_a$ values in bulk electrolyte and at interphases composed of halide salts (see Supplementary Table 1) indicates that there is only a modest change in the barrier for ion movement as ions leave the bulk electrolyte and cross the electrolyte-electrode interface during deposition, which would be expected to favor more stable deposition. This inference is confirmed by the fact that the difference in energy barrier is lowest for Mg cells, which do not form dendrites.

**Electrodeposition of sodium metal with NaBr-coated anode.** It is known that sodium metal is more reactive than lithium[6], thus in contact with a liquid electrolyte it is expected to fail more easily by the first of the three mechanisms discussed in the introduction. Protecting the Na surface with a coating like NaBr that does not increase the barrier for Na ion transport at the interface should therefore inhibit failure by this mechanism. Additionally, since rough deposition is triggered by formation of an inhomogeneous SEI, cell failure by dendrite-induced short circuits should also be lower. To evaluate these statements, we directly visualize the surface of the Na anode with/without NaBr coatings during electrodeposition and quantify the growth of surface roughness as

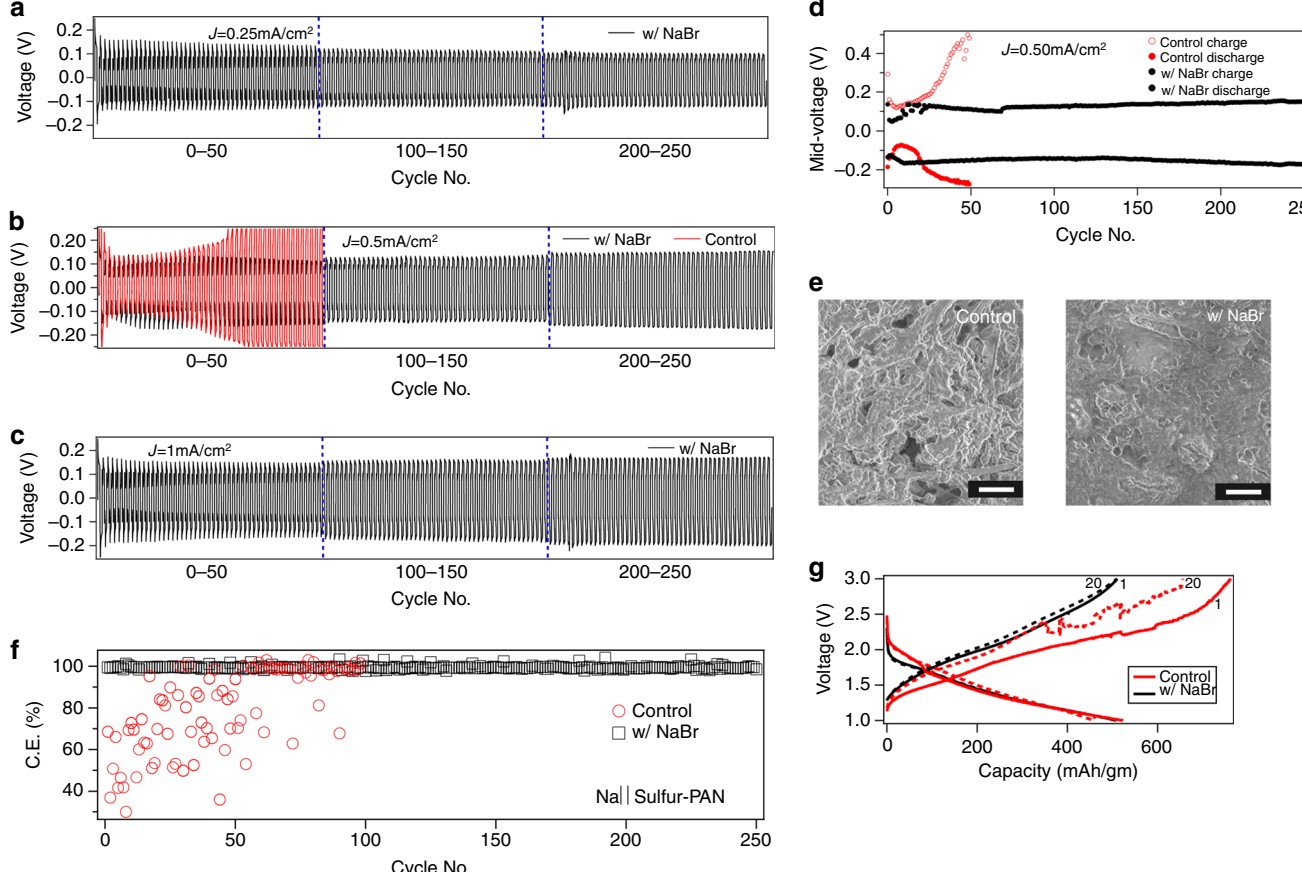

**Fig. 6** Galvanostatic cycling performance of Na anodes. Voltage profile obtained by consecutively charging and discharging a symmetric sodium cell at current densities of **a** 0.25 mA cm$^{-2}$, **b** 0.5 mA cm$^{-2}$, **c** 1 mA cm$^{-2}$. In each experiment, one complete cycle was 1 h long, with each charge and discharge time is half-an-hour. The profiles in *black* represent sodium metals with NaBr coating and *red* stands for pristine sodium (control). **d** The voltage hysteresis represented by the mid-voltage values of charge and discharge is plotted as a function of cycle no. for NaBr coated and control sodium cells corresponding to (**b**). **e** Morphology of sodium metal electrode obtained from post-mortem SEM analysis. The cells were charged at a rate of 1 mA cm$^{-2}$ for 2 h before being taken apart for the post-mortem analysis. The *left* image are results for pristine Na, while the image to the *right* is for NaBr-coated Na; for both images: *scale bar*, 5 μm. **f** Coulombic efficiency of a Na||Sulfur-PAN composite half-cell as a function of cycle number. **g** Voltage profile of Na||Sulfur-PAN during charging and discharging at the 1st and 20th cycle numbers. The *red lines/symbols* represent control, while *black* shows the result for NaBr-coated sodium

a function of time. A symmetric cuvette-type optical cell (see Supplementary Fig. 6) was used for this component of the study. In a typical experiment, the cell is polarized with a fixed current density of $1\,mA\,cm^{-2}$ and the morphology of the electrode-surface viewed in an up-right Olympus optical microscope outfitted with 6.10 mm extra-long working distance 10X objectives. Videos of the visualization-experiment are provided (with a 300X speed). Figure 5a reports images obtained from these measurements at discrete time points separated by a fixed 10-min interval. It is observed that a pristine Na electrode is prone to form moss-like dendritic structures. In comparison, the NaBr-coated sodium metal is seen to electrodeposit without the formation any dendritic structure; however, it is seen that the overall volume (thickness) of the electrode is increasing over time due to the deposition under the NaBr coating. Figure 5b plots the tip length of the dendrites over time. It is seen that for the pristine Na electrode, dendrites begin to grow immediately upon imposition of the current and grow throughout the electrode surface. The number density and reactivity of dendrites can be estimated from the brightness of the respective images. Reaction of Na with electrolyte is known to cause the metal to lose its shiny appearance. This, along with the much greater number density of dendritic structures is the source of the darker appearance of the images obtained using the pristine Na electrode. Figure 5c reports the comparative Lightness (L*) of different spots on the sodium metal for both cases. Here, Lightness (L*) is defined as the relative brightness of a spot, such that a white spot would correspond L* of 100 and that of a black spot would be zero. L* of pristine sodium drops down close to zero within 1 min of electrodeposition, while that of sodium with NaBr maintains L* > 90 for entire time of measurement. It is observed that the reduction in brightness (or synonymously formation of undesired by-products) is synchronous to the growth of dendrite-like structure. It can be hypothesized that the distribution of the insulating products is heterogeneous, which leads to a non-uniformity of local current density on the electrode surface that ultimately causes the formation of rough and needle-like structures. These dendrites further increment the local current densities due to their sharp edges causing a cascade of instabilities.

Figure 6 reports the electrochemical performance of sodium-metal cells in different configurations. To simulate the performance of a working Na metal battery, a symmetric Na coin cell was cycled galvanostatically at various fixed current densities of $0.25\,mA\,cm^{-2}$, $0.5\,mA\,cm^{-2}$ and $1.0\,mA\,cm^{-2}$ and the voltage profile is reported as a function of time in Fig. 6a, b and c, respectively. For all the different current density values, the voltage profiles are flat. The voltage hysteresis, of the sodium metal plating and stripping, represented by the mid-voltage value of charge-and-discharge, is plotted in Fig. 6d. On comparing results for pristine and NaBr-coated Na electrodes (in Fig. 6b, d), it is seen that for cell with pristine Na, the voltage diverges to > 0.5 V after 30 h of cycling, meaning that the effective interfacial resistance is > 15kΩ. In contrast, the cell comprising NaBr-coated Na is stable for at least 250 h with minimal rise in cell voltage and hence effective resistance. The divergence in cell resistance is most likely a result of formation of a thick and insulating SEI layer; such an observation with pristine sodium anodes complements the result obtained from visualization experiments explained in Fig. 5. However, it is remarkable that with the NaBr coating the overpotential is nearly constant for 250 h, indicating that the NaBr coating produces nearly a 10-fold improvement in the cell lifetime.

Post-mortem analysis of polarized sodium anodes was done using SEM after continuously electrodepositing at the rate of $1\,mA\,cm^{-2}$ for 2 h. Figure 6e shows the morphology of the sodium metal with and without NaBr coating. It is seen that the pristine Na electrode develops a non-uniform surface with few protruding sharp structures compared to a relatively smooth surface for the NaBr-coated electrode. It is important to note that in contrast to visualization experiments the general electrodeposition is more uniform, which can be attributed to the stabilizing effect of compression forces exerted by the separator on Na electrodes in the coin cells.

Finally, the effectiveness of the NaBr-coated Na metal was evaluated in a full Na||S electrochemical cell, comprising a sulfur-polyacrylonitrile composite (SPAN) cathode. In this cathode, molecular sulfur is covalently trapped in a PAN framework, which has been reported to completely eliminate polysulfide dissolution and shuttling effects with carbonate-based solvents in lithium-sulfur cells[52]. However, unlike their Li counterparts, the anodes in Na||SPAN cells have been reported to develop "black mossy dendrites" within a few cycles, which results in an unstable voltage profile during the cell recharge and low coulombic efficiency[53]. This effect is apparent in the *inset* to Fig. 6g for the Na||SPAN cells based on pristine Na anodes. The lower coulombic efficiency measured in Na||SPAN cells employing pristine Na as anode is also observed from Fig. 6f. Because the reactivity of sodium with electrolytes increases with voltage, we conclude that Na||SPAN cell configuration provides a more rigorous assessment of electrode reactivity and stability, in comparison to the Na||Cu type cells[18] used in previous work. Here we observe that coating Na with NaBr protects the metal and in a liquid electrolyte, and in the presence of a conversion cathode, yields columbic efficiency > 99% for at least 250 cycles with minimal fade in discharge capacity as plotted in Supplementary Fig. 7a. Also, the SEM image of the Na anode (with NaBr protection) is shown in Supplementary Fig. 7b along with the EDX mapping of elements. No dendritic features are seen even after the long term cycling, and in addition the sulfur species are absent, indicating immunity of the anode from side reactions. This observation can be attributed to the near complete protection of the metal from parasitic reactions, without compromising ion transport across the SEI. Thus, we conclude that consistent with the JDFT calculations a NaBr-coated Na anode opens the possibility of stable, room-temperature rechargeable sodium metal batteries able to operate using liquid electrolytes.

## Discussion

We used DFT calculations to analyze the surface diffusion barriers for Na and Li adatom transport on various salts. It was observed that the usually formed SEI components on Li, including LiOH and $Li_2CO_3$, have very high activation energy barriers, which is thought to increase the propensity of the metal to form needle-like, dendrite nucleates. In contrast, energy barriers for adatom diffusion on metal halide salts, including LiF, NaF, NaBr, are low, with the energy barrier for diffusion on NaBr as low as that of Magnesium, which is known to form spherical nucleates on charging. We evaluate these predictions using Na electrodes, on which an artificial SEI composed of pure NaBr is used to protect the metal. By means of XRD, EDAX, XPS, and cryo-FIB-SEM measurements, we confirm that NaBr coatings with thicknesses ranging from 2 μm–12 μm are achieved on Na. Further, impedance spectroscopy measurements at different temperatures show that NaBr-coated sodium anodes exhibit at least three times lesser interfacial ion-transport activation energy compared to pristine sodium. In-situ visualization was performed to contrast the electrodeposition-stability with and without NaBr layer on sodium anodes. It showed that the NaBr coating not only restricts dendritic formation, but also prevents unwanted side-reactions between the electrode and electrolyte. This observation

is in line with the charge-discharge measurements in symmetric cells as well as in coulombic measurement tests in Na||SPAN type half-cells. Thus, we think that this rational analysis of SEI layers in reactive metal-batteries, as well as the methodology of incorporating the desired component in the SEI, can provide a new outlook towards low-cost and long lasting secondary batteries.

## Methods

**Materials**. Sodium cubes, bromo-propane, chloro-propane, iodo-propane, propylene carbonate (PC), ethylene carbonate (EC), diglyme, dimethoxyethane, sodium hexafluorophosphate, magnesium(II) bis(trifluoromethanesulfonyl)imide, were all purchased from Sigma Aldrich. Celgard 3501 separator was obtained from Celgard Inc. Glass fiber separator was bought from Whatman Inc. All the chemicals were used as received in after rigorous drying in a ~ 0 ppm water level and <5 ppm oxygen glove box; in order to make sure the sodium metal is not oxidized.

**Sodium bromide and other halide coating formation**. Sodium-cube pieces were taken out of mineral oil and cleaned with kimwipes. Then with a sharp knife, thin slices of sodium pieces were cut before punching with ¼th inch punch. For the coating of solid electrolyte interface, 15 μl of bromo-propane was added to the sodium electrode before drying in vacuum ante-chamber for five minutes. It is known that the reaction is instantaneous due to high reactivity of sodium metal. Further the by-product obtained by reaction- hexane is believed to vaporize rapidly in vacuum owing to its low boiling point characteristics. Coating with NaCl and NaI was done in exact same procedure, however, with chloro-propane and iodo-propane respectively.

**Physical characterization**. XPS was conducted using Surface Science Instruments SSX-100 with operating pressure of ~ $2 \times 10^{-9}$ torr. Monochromatic Al K-α x-rays (1486.6 eV) with beam diameter of 1mm were used. Photoelectrons were collected at an emission angle of 55°. A hemispherical analyzer determined electron kinetic energy, using pass energy of 150 V for wide survey scans and 50 V for high-resolution scans. Samples were ion-etched using 4 kV Ar ions, which were rastered over an area of $2.25 \times 4$mm with total ion beam current of 2 mA, to remove adventitious carbon. Depth profile was obtained was obtained by ion etching at 2 kV, 22 μA over 2*3 mm, which yielded an atch rate of approximately 5 nm min$^{-1}$. The etching was done for 395 min. Spectra were referenced to adventitious C 1 s at 284.5 eV. CasaXPS software was used for the XPS data analysis with Shelby backgrounds. Br 3$d$ was assigned to double peaks (3$d_{5/2}$ and 3$d_{3/2}$) for each bond with 1.05 eV separation. Residual SD was maintained close to 1.0 for the calculated fits. Samples were exposed to air only during the short transfer time to the XPS chamber (less than 5 s).

XRD was carried out on a Scintag Theta-Theta X-ray diffractometer using Cu K$_\alpha$ radiation at $\lambda = 1.5406$ Å. All samples were covered with Kapton tape to ensure that the sodium metal is not oxidized in air. For the sodium metal samples after cycling, the symmetric cell of NaBr-coated sodium was charged-and-discharged five times at a current density of 0.5 mA cm$^{-2}$.

**Electrochemical characterization**. The impedance spectroscopy measurement ionic conductivity of electrolytes was measured as a function of temperature using a Novocontrol N40 Broadband Dielectric instrument. Symmetric cells were prepared using two sodium pristine metal pieces using the electrolyte 1 M NaPF$_6$ EC/PC with a glass fiber separator. For understanding the impedance of halide-based interfacial layer, both sodium pieces were coated with respective halides using the same method (as described for NaBr-coating section) before performing the experiment. For the symmetric cell with Mg electrodes, the electrolyte 0.3 M Mg (TFSI)$_2$ in DME/diglyme was used with a glass fiber separator. The measurements were done in a frequency range from 10$^{-3}$ to 10$^7$ Hz.

**Scanning electron microscopy**. Post-mortem characterization of the sodium metal electrodes was done to understand the morphology of sodium metal deposition and also failure mechanisms involved in short circuits. For this reason, the cells with symmetric sodium cells with or without NaBr layer was charged at a current density of 1 mA cm$^{-2}$ for 2 h before disassembling the cell inside the glovebox. The charged sodium metal pieces were washed with the electrolyte-solvent and were transferred carefully to the microscopy facility minimizing the exposure of sodium metal to atmosphere. The SEM analysis was done using the LEO155FESEM instrument.

**Focused ion beam/SEM**. An FEI Strata 400 Focused Ion Beam (FIB) was used for the FIB/SEM experiments. The FIB is fitted with a Quorum PP3010T Cryo-FIB/SEM Preparation System that enables cryogenic experiments. For room temperature experiments, the sample was removed from an inert environment and loaded onto an aluminum stub. The stub was subsequently attached to a shuttle and placed in a loadlock that pumps down to vacuum before inserting into the FIB. For cryogenic experiments, the sample was removed from an inert environment, attached to a stub, and immediately plunged into slush nitrogen. This shortened

exposure time and minimized any reactions with air or moisture. The sample was then transferred into the FIB in a transfer device at liquid nitrogen temperature and subsequently maintained at −165 °C in the cryo-FIB.

**In situ visualization studies**. The visualization experiment was carried out for understanding the in-operando observation of electrodeposition in sodium metal batteries. In all experiments the electrolyte- 1 M NaPF$_6$- EC/PC was used. For control experiments pristine sodium was used as both electrodes, while for understanding the role of stable SEI, NaBr-coated sodium pieces were used. The sodium metal pieces were attached to current collectors and fixed in an air-tight cuvette-chamber as shown in Supplementary Fig. 3. In these experiment, it is made sure that the electrodes are fully facing each other and then one electrode is continuously charged with a current density of 1 mA cm$^{-2}$. The electrode, being charged was monitored over time and images of sodium deposition at different intervals were captured from an optical microscope.

**Cell lifetime and failure studies**. Symmetric 2032 type Na|Na coin cells with and without NaBr coating containing liquid electrolyte of 1 M NaPF$_6$ EC/PC (1:1 v/v) inside an argon-filled glove box. The cells were evaluated using galvanostatic (strip-plate) cycling using a Neware CT-3008 battery tester. In the 'strip-plate' experiments, the batteries were repeatedly charged and discharged with each half-cycle 0.5 h long. Failure was deduced from irregularities in the voltage profile as well as excessive increment in the overpotential indicating excessive formation of electrolyte by-products completely insulating the electrodes.

**Sulfur-PAN cathode cycling**. Galvanostatic measurements with the Sulfur-PAN composite cathode was done in a 2032-type coin cell comprising a sodium metal anode with and without the NaBr coating. Celgard 3501 polypropylene membranes were used as the separator. 40 μL 1 M NaClO$_4$ in a mixture EC and PC (v:v = 1:1) was used as the electrolyte. The cathode consisted of 70 wt% of the active material, 15 wt% of carbon black (Super-P Li from TIMCAL) as a conductivity aid, and 15 wt% of polymer binder (PVDF, polyvinylidene fluoride, Aldrich). A carbon-coated aluminum foil used for current collector. The mass loading of the cathode was ~ 0.85 mg SPAN cm$^{-2}$. Detailed synthesis of the PAN-Sulfur composite can be found in recent paper by Wei et al.[52] Cell assembly was carried out in an argon-filled glove-box. The measurements were done in Neware CT-3008 battery tester.

**Data availability**. The authors declare that the data supporting the findings of this study are available within the paper and its Supplementation Information files or from the corresponding author upon reasonable request.

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

## Acknowledgements

This work was supported by the Department of Energy, Advanced Research Projects Agency—Energy (ARPA-E) through award #DE-AR0000750. The work made use of electrochemical characterization facilities in the KAUST-CU Center for Energy and Sustainability, supported by the King Abdullah University of Science and Technology (KAUST) through Award # KUS-C1-018-02. Electron microscopy facilities at the Cornell Center for Materials Research (CCMR), an NSF-supported MRSEC through Grant DMR-1120296, were also used for the study. M.J.Z. and L.F.K. acknowledge support by the NSF (DMR-1654596).

## Author contributions

S.C. and S.W.: Contributed equally to the paper. S.C. and L.A.A.: Conceived the idea. S.C., S.W., L.A.A.: Conducted the electrochemical measurements and analyzed the data. Y.O., D.G., T.A.A.: Performed the J-DFT calculations. M.J.Z. and L.F.K.: Performed cryo-SEM and EDX analysis. Z.T., S.W., S.C.: Performed XPS and SEM measurements. J.H.S. and S.C.: Conducted impedance spectroscopy measurements. S.W., P.N., S.C., A.A.: Performed the visualizations studies and analysis. All authors contributed to the writing of this article.

## Additional information

**Competing interests:** The authors declare no competing financial interests.

