## [Peer Review File · Nature Communications]

Reviewers' comments:

Reviewer #1 (Remarks to the Author):

Controlling the transport of Na ions on the surface of Na metal and suppressing the growth of Na dendrites are vital in improving the electrochemical performance of Na metal batteries. This manuscript analyzes the surface diffusion barriers of different compounds of lithium and sodium by means of Density Functional Theory calculations and found that NaBr has very low surface diffusion barrier for interfacial ion transport. Experimentally, the authors show that a protective layer of NaBr helps to stabilize the Na metal and improve the electrochemical performances. This is a crisp study on the design of a stable artificial SEI for improving the Na metal batteries. This work is of broad enough interest that it will attract the battery community. However, the manuscript consists of some flaws that should be clarified. Nature Communications is a flagship journal. This work can be published in it after a major revision. Below is a list of the questions.

1. More detailed information of the as-formed NaBr film should be provided, such as the thickness of the film and the particle size of NaBr. Because the thickness and the particle size of film have significant influence on both the mechanical properties of the film and the battery performance.
2. The SEM image of NaBr modified Na metal is not clear (Figure 2c).
3. Although Na dendrites do not grow out in the in situ visualization studies due to the protection of the NaBr layer, the situation might change in a real cell using the NaBr modified Na metal. Na metal is very soft, while NaBr film is rigid. The authors should investigate whether the NaBr layer on Na metal cracks or not under mechanical pressure during battery assembling?
4. Page 6 Line 10. The authors state that "The presence of characteristic peaks of NaBr crystals in the coated sodium metal confirms the success of the reaction in uniformly depositing NaBr on the electrode surface." The XRD result can only confirm the existence of the NaBr but cannot verify that whether the deposition is uniform.
5. Page 6 Line 11. The authors state that "In addition, the XRD results for sodium metal anode (w/ NaBr coating) was obtained after cycling five times in a symmetric sodium battery with 0.5mA/cm². As shown in Figure S1 of Supplementary Information, the NaBr coating is retained, confirming that the anode-protection mechanism is sustained even after the even after rigorous galvanostatic cycling." Again, the XRD result can only indicate that NaBr is electrochemically stable but cannot tell whether the NaBr film is well retained without cracks (Figure S1). More detailed information, such as SEM images after cycling, should be provided to support the authors' claim.
6. There is no clear description on the voltage hysteresis of Na metal plating and stripping for the bare Na metal and the modified Na metal. In Figure 5, the detailed voltage profiles at the initial cycle and after many cycles should be provided for the bare Na metal and the NaBr modified Na metal.

Reviewer #2 (Remarks to the Author):

Na metal, as the promising anode materials for Na metal batteries, attract increasing attention recently. However, similar with Li anode, Na anode also suffers from the serious problem, such as dendrite growth and high reactivity with electrolyte. This manuscript reports the sodium bromide coating on Na metal with low energy barrier to achieve the dendrite free Na anode. It is very interesting to first show the Joint Density Functional Theoretical analysis in Na protection areas and improved life time of Na metal anode. However, the electrochemical experiments were not rigorously designed and therefore their conclusions may not be consistent with the results. Therefore, the present work is not appropriate to be published in "Nature communication". They should at least add more electrochemical experiments and detailed information to support their conclusion. The detailed comments are listed below:

- 1) What is the thickness of NaBr coating on the surface of Na metal? Does the thickness of coating layer affect the electrochemical performances?
- 2) How is the performance such as over-potential and cycle stability different according to the current rate? Broader range of current rate result should to be shown. It is insufficient to show the NaBr coating effect at only 0.5 mAcm⁻².
- 3) The potential-capacity voltage profiles are suggested to be shown.
- 4) It is claimed that the NaBr plays the role of artificial SEI on Na surface. How is the SEI formation on the Na surface after electrochemical cycling? What is the difference between pristine Na and NaBr-coated Na on the SEI formation? More characterizations like depth profiling XPS, SIMS, or Raman are suggested to be added. Only XRD is insufficient.
- 5) It is very promising that there is obvious difference of Coulombic efficiency in Na-S half-cell. However, what is the surface change for both morphologies and chemical state on Na anode? It is also suggested to be analyzed.

Reviewers' comments:

Reviewer #1 (Remarks to the Author):

I have carefully read the replies of the authors to my original comments and the authors fully addressed my comments. I think that the revised paper can be published.

Reviewer #3 (Remarks to the Author):

The application of the sodium anode, similar with lithium anode, is hindered by serious safety concerns caused by the uneven deposition of Na and the growth of Na dendrites. In order to developing rational strategies for stabilizing the anode of Na batteries, this manuscript investigate the surface diffusion barriers for Na and Li adatom transport on various salts by Joint Density Functional Theory (JDFT). And they found that NaBr has the lowest diffusion barrier for interfacial ion transport. Following, electrochemical experiments were carried out to evaluate the JDFT prediction. And they show that the NaBr coating-layer on a Na metal electrode restrict dendritic formations and prevent the unwanted side-reactions between the electrode and electrolyte. This work is very interesting, but there are some flaws. I recommend the publication of this work after well addressing the following issues.

1. There is no clear description on the electrodeposit mechanism of the sodium metal anode (with NaBr coating layer). To start with, page 9 line 253, the authors state that "In comparison, the NaBr coated sodium metal is seen to electrodeposit without the formation any dendritic structure, however it is seen that the overall volume (thickness) of the electrode is increasing over time due to the deposition under the NaBr coating (see Supplementary Information video)." Sodium metal is deposition under the NaBr coating, that is to say, on the surface of Na-metal. What's the behavior of the deposited Na under the NaBr coating layer. What the effect of the NaBr coating layer? Is it a barrier layer or not? The morphology of the NaBr layer (Figure 2(c)) clearly shows two regions of a smooth deposit and a porous regions on Na. If it is the barrier layer on the deposited Na, the chemical and modulus properties of the two region need to be study carefully. On the barrier mechanism, the authors may glad to know there are two related papers referring to dendrite suppression (Adv. Mater. 2016, 28, 1853–1858; Nano Energy 2017, 36, 411–417), which should be cited and discussed on their common points.
2. The snapshots from video microscopy of NaBr coated Na-electrolyte interface (Figure 5(a) right) are not clear.
3. According to the results of JDFT, NaBr has the lowest diffusion barrier for interfacial ion transport, when compared with other sodium halide salts and lithium salts. More comparison such as NaF or NaI are suggested to be shown.
4. In the symmetric Na coin cell, the electrodepositing time for each cycle at current densities of 0.25mA/cm², 0.5mA/cm² and 1.0mA/cm². And the zoom in voltage profile at the three current densities are suggested.
5. Page 10 Line 302. The authors state that "In this cathode, molecular sulfur is covalently trapped in a PAN framework, which has been reported to completely eliminate polysulfide dissolution and shuttling effects with carbonate based solvents in lithium-sulfur cells." However, the control Na||SPAN cells are hard to understand. First, in voltage profile of Figure 6g, the charge capacity are much higher than discharge capacity which is opposite with the common phenomenon in sodium-sulfur cells. Please explain why. Secondly, the Coulombic efficiency of the control cells are extremely bad, when compared with many publications in sodium-SPAN and lithium-SPAN (such as ref 49 of the manuscript). More control experiments maybe needed or more reasons should be given.

REVIEWERS' COMMENTS:

Reviewer #3 (Remarks to the Author):

Paper is acceptable as is.

Reviewer #1 (Remarks to the Author):

Controlling the transport of Na ions on the surface of Na metal and suppressing the growth of Na dendrites are vital in improving the electrochemical performance of Na metal batteries. This manuscript analyzes the surface diffusion barriers of different compounds of lithium and sodium by means of Density Functional Theory calculations and found that NaBr has very low surface diffusion barrier for interfacial ion transport. Experimentally, the authors show that a protective layer of NaBr helps to stabilize the Na metal and improve the electrochemical performances. This is a crisp study on the design of a stable artificial SEI for improving the Na metal batteries. This work is of broad enough interest that it will attract the battery community. However, the manuscript consists of some flaws that should be clarified. Nature Communications is a flagship journal. This work can be published in it after a major revision. Below is a list of the questions.

1. More detailed information of the as-formed NaBr film should be provided, such as the thickness of the film and the particle size of NaBr. Because the thickness and the particle size of film have significant influence on both the mechanical properties of the film and the battery performance.

Response: The reviewer's point is well taken. We've now more fully analyzed the NaBr-modified sodium metal anode and the results are discussed on pgs 5 and 6 of the revised paper. Specifically, we've used cryo-focused ion beam-scanning electron microscopy (cryo-FIB-SEM) to perform cross-sectional images of the Na electrodes after 1 and 5 mins of treatment with bromopropane. The results show that compact and uniform surface coatings with thicknesses of 2 μ m and 12 μ m, respectively are achieved. Depth-dependent composition analysis of the coating layer determined using energy dispersive x-ray (EDX) mapping of cross sections produced by FIB milling and XPS depth profiling show that the coating is comprised of NaBr.

2. The SEM image of NaBr modified Na metal is not clear (Figure 2c).

Response: The figure has been replaced in the revised manuscript.

3. Although Na dendrites do not grow out in the in situ visualization studies due to the protection of the NaBr layer, the situation might change in a real cell using the NaBr modified Na metal. Na metal is very soft, while NaBr film is rigid. The authors should investigate whether the NaBr layer on Na metal cracks or not under mechanical pressure during battery assembling?

Response: Prompted by the referee's suggestion, we performed postmortem XPS depth profiling on 2x3mm Na electrode segments after cycling and confirmed that the NaBr layer remains largely intact after battery assembly and cycling.

4. Page 6 Line 10. The authors state that "The presence of characteristic peaks of NaBr crystals in the coated sodium metal confirms the success of the reaction in uniformly depositing NaBr on the electrode surface." The XRD result can only confirm the existence of the NaBr but cannot verify that whether the deposition is uniform.

Response: The reviewer makes a valid point. The above line has been modified in the revised manuscript. Additional survey SEM and cryo-FIB/SEM were performed to investigate the uniformity of the NaBr coatings on Na metal. These results are discussed on pages 5 and 6 of the revised manuscript.

5. Page 6 Line 11. The authors state that "In addition, the XRD results for sodium metal anode

(w/ NaBr coating) was obtained after cycling five times in a symmetric sodium battery with 0.5mA/cm². As shown in Figure S1 of Supplementary Information, the NaBr coating is retained, confirming that the anode-protection mechanism is sustained even after the even after rigorous galvanostatic cycling.” Again, the XRD result can only indicate that NaBr is electrochemically stable but cannot tell whether the NaBr film is well retained without cracks (Figure S1). More detailed information, such as SEM images after cycling, should be provided to support the authors’ claim.

Response: We supplemented these measurements with more in-dept analyses, including SEM, EDX, and XPS depth profiling analyses and the results are provided in Figure 3.

6. There is no clear description on the voltage hysteresis of Na metal plating and stripping for the bare Na metal and the modified Na metal. In Figure 5, the detailed voltage profiles at the initial cycle and after many cycles should be provided for the bare Na metal and the NaBr modified Na metal.

Response: The hysteresis of Na plating and stripping for NaBr protected and control cells are provided in Figure 6(d).

Reviewer #2 (Remarks to the Author):

Na metal, as the promising anode materials for Na metal batteries, attract increasing attention recently. However, similar with Li anode, Na anode also suffers from the serious problem, such as dendrite growth and high reactivity with electrolyte. This manuscript reports the sodium bromide coating on Na metal with low energy barrier to achieve the dendrite free Na anode. It is very interesting to first show the Joint Density Functional Theoretical analysis in Na protection areas and improved life time of Na metal anode. However, the electrochemical experiments were not rigorously designed and therefore their conclusions may not be consistent with the results. Therefore, the present work is not appropriate to be published in “Nature communication”. They should at least add more electrochemical experiments and detailed information to support their conclusion. The detailed comments are listed below:

1) What is the thickness of NaBr coating on the surface of Na metal? Does the thickness of coating layer affect the electrochemical performances?

Response: Prompted by the reviewer question, we characterized the thickness of NaBr coating using cross-sectional cryo-focused ion beam-scanning electron microscopy (cryo-FIB-SEM) after 1 and 5 mins of treatment with bromopropane. The results are discussed on pages 5 and 6 of the revised manuscript. Briefly, we find that compact and uniform surface coatings with thicknesses of 2 μ m and 12 μ m, respectively are achieved. Depth-dependent composition analysis of the coating layer determined using energy dispersive x-ray (EDX) mapping of cross sections produced by FIB milling and XPS depth profiling show that the coating is comprised of NaBr.

2) How is the performance such as over-potential and cycle stability different according to the current rate? Broader range of current rate result should to be shown. It is insufficient to show the NaBr coating effect at only 0.5 mAcm⁻².

Response: We agree. The hysteresis of Na metal plating and stripping w/ and w/o NaBr coating is shown in figure 6(d). Symmetric cell cycling data at current densities of 0.25mA/cm² and 1mA/cm² are also provided.

3) The potential-capacity voltage profiles are suggested to be shown.

Response: Done. The potential-capacity profiles are shown in Figure 6(g).

4) It is claimed that the NaBr plays the role of artificial SEI on Na surface. How is the SEI formation on the Na surface after electrochemical cycling? What is the difference between pristine Na and NaBr-coated Na on the SEI formation? More characterizations like depth profiling XPS, SIMS, or Raman are suggested to be added. Only XRD is insufficient.

Response: Agreed. The depth profiling result with XPS after cycling is shown in Figure 3(c,d) and the SEM image along with EDX is represented in Figure 3(b,e).

5) It is very promising that there is obvious difference of Coulombic efficiency in Na-S half-cell. However, what is the surface change for both morphologies and chemical state on Na anode? It is also suggested to be analyzed.

Response: This is a good suggestion. The requested surface morphology of Na anode after cycling with NaBr coating along with EDX are provided in Figure S6 (b).

Reviewer #3 (Remarks to the Author):

The application of the sodium anode, similar with lithium anode, is hindered by serious safety concerns caused by the uneven deposition of Na and the growth of Na dendrites. In order to developing rational strategies for stabilizing the anode of Na batteries, this manuscript investigate the surface diffusion barriers for Na and Li adatom transport on various salts by Joint Density Functional Theory (JDFT). And they found that NaBr has the lowest diffusion barrier for interfacial ion transport. Following, electrochemical experiments were carried out to evaluate the JDFT prediction. And they show that the NaBr coating-layer on a Na metal electrode restrict dendritic formations and prevent the unwanted side-reactions between the electrode and electrolyte. This work is very interesting, but there are some flaws. I recommend the publication of this work after well addressing the following issues.

1. There is no clear description on the electrodeposit mechanism of the sodium metal anode (with NaBr coating layer). To start with, page 9 line 253, the authors state that “In comparison, the NaBr coated sodium metal is seen to electrodeposit without the formation any dendritic structure, however it is seen that the overall volume (thickness) of the electrode is increasing over time due to the deposition under the NaBr coating (see Supplementary Information video).” Sodium metal is deposition under the NaBr coating, that is to say, on the surface of Na-metal. What’s the behavior of the deposited Na under the NaBr coating layer. What the effect of the NaBr coating layer? Is it a barrier layer or not? The morphology of the NaBr layer (Figure 2(c)) clearly shows two regions of a smooth deposit and a porous regions on Na. If it is the barrier layer on the deposited Na, the chemical and modulus properties of the two region need to be study carefully. On the barrier mechanism, the authors may glad to know there are two related papers referring to dendrite suppression (Adv. Mater. 2016, 28, 1853–1858; Nano Energy 2017, 36, 411–417), which should be cited and discussed on their common points.

Response: We thank the reviewer for pointing out these relevant papers and have included both works in the revised manuscript.

NaBr is electronically insulating, sodium ions are therefore thought to be transported through the coatings and deposit under the NaBr layer. The uniformity of the deposited layer is facilitated both by the by the fast surface diffusion of Na ions in the NaBr layer and protection the coatings offer against side reactions with the electrolyte solvent. In response to the reviewer’s comment about the differences in the properties of the NaBr coated sodium versus pristine sodium, we consider results from an aggressive air exposure test designed to directly illustrate that even regions on the SEM where the NaBr layer appears thinner are protected. Specifically, the NaBr-coated Na and pristine Na were each exposed for 30 seconds to room air and this exposure was followed by cross-sectional SEM analysis using Focussed Ion Beam milling at ambient conditions. As seen in Figure 3-A,B there are significant and obvious differences throughout the $33\mu\text{m} \times 33\mu\text{m}$ specimen area. It is observed for example that while the pristine Na shows evidence of severe air and moisture-induced corrosion, the NaBr coated Na metal remains smooth and well protected over the entire specimen area.

2. The snapshots from video microscopy of NaBr coated Na-electrolyte interface (Figure 5(a) right) are not clear.

Response: Clearer and high quality images from the video are provided in the revised manuscript.

3. According to the results of JDFT, NaBr has the lowest diffusion barrier for interfacial ion transport, when compared with other sodium halide salts and lithium salts. More comparison such as NaF or NaI are suggested to be shown.

Response: The effectiveness of other sodium halides was studied by temperature-dependent EIS and the results reported in Figure 4. NaBr coated sodium is seen to have the lowest interfacial activation energy compared to other halide-coated sodium metal in agreement with the JDFT results. Previously, there have been reports of indirectly generating NaF in the SEI layer utilizing 1M NaPF₆ in diglyme (ACS Cent. Sci., 2015, 1 (8), pp 449–455) or FEC additives (ACS Appl. Mater. Interfaces, 2011, 3 (11), pp 4165–4168), which also exhibit improved cycling performance compared to control sodium anodes.

4. In the symmetric Na coin cell, the electrodepositing time for each cycle at current densities of 0.25mA/cm², 0.5mA/cm² and 1.0mA/cm². And the zoom in voltage profile at the three current densities are suggested.

Response: The electrodeposition time for each cycle was maintained at 1 hour long. This information is provided in the figure caption. The zoomed-in voltage profile is also provided in the revised manuscript.

5. Page 10 Line 302. The authors state that “In this cathode, molecular sulfur is covalently trapped in a PAN framework, which has been reported to completely eliminate polysulfide dissolution and shuttling effects with carbonate based solvents in lithium-sulfur cells.” However, the control Na||SPAN cells are hard to understand. First, in voltage profile of Figure 6g, the charge capacity are much higher than discharge capacity which is opposite with the common phenomenon in sodium-sulfur cells. Please explain why. Secondly, the Coulombic efficiency of the control cells are extremely bad, when compared with many publications in sodium-SPAN and lithium-SPAN (such as ref 49 of the manuscript). More control experiments maybe needed or more reasons should be given.

Response: The long charge capacity compared to discharge capacity observed for the controls is associated with side reactions between mossy sodium metal deposits and electrolyte components; the low coulombic efficiency is another signature of such instabilities. Previously, Wang et al. (<https://doi.org/10.1016/j.elecom.2006.08.029>) reported on Na||SPAN cycling for only 20 cycles, thereafter the authors reported severe degradation of sodium anode. Quoting from the paper, “After opening the battery in an argon-filled glove bag, it was found that thick black moss, so-called “sodium dendrite”, covered the surface of the sodium anode.” The control result for Na||SPAN battery is indeed quite bad, but robust; results from all of the studied cells (see example below) show noisy and nearly identically low Coulombic efficiencies.